# A study of Hate Speech in Social Media during the COVID-19 outbreak

**Viviana Cotik(1)**      **Natalia Debandi (2)**      **Franco Luque (3)**
**Paula Miguel (5)**      **Agustín Moro (4)**      **Juan Manuel Pérez (1)**
**Pablo Serrati (5)**      **Joaquin Zajac (5)**      **Demián Zayat (6)**

(1) DC, FCEyN, Universidad de Buenos Aires & ICC-CONICET
(2) Universidad Nacional de Río Negro. IIPPyG-CONICET
(3) Universidad Nacional de Córdoba & CONICET
(4) Universidad de Buenos Aires & Universidad Nacional del Centro
(5) IIGG, FSOC, Universidad de Buenos Aires & CONICET
(6) INADI & Universidad de Buenos Aires
vcotik@dc.uba.ar

## Abstract

In pandemic situations, hate speech propagates in social media, new forms of stigmatization arise and new groups are targeted with this kind of speech.

In this short article, we present work in progress on the study of hate speech in Spanish tweets related to newspaper articles about the COVID-19 pandemic.

We cover two main aspects: The construction of a new corpus annotated for hate speech in Spanish tweets, and the analysis of the collected data to answer questions from the social field, aided by modern computational tools.

Definitions and progress are presented in both aspects. For the corpus, we introduce the data collection process, the annotation schema and criteria, and the data statement. For the analysis, we present our goals and their associated questions. We also describe the definition and training of a hate speech classifier and present preliminary results using it.

## 1 Introduction

Hate speech is pervasive in social media and an increasingly worrying issue in society. Historically discriminated communities are a frequent target of prejudice, insult, offense, and even call for violent actions, with damaging consequences. In pandemic situations like the current COVID-19 outbreak, social and mass media consumption is intensified, turning social networks into one of the main scenarios where hate speech propagates. In this particular context, new forms of stigmatization arises and new groups are targeted with this kind of speech.

Research on hate speech detection has been considerably active in the last years. Most work is focused on short user generated content (such as tweets) rarely taking context into account. However, context is relevant to address user intention in short texts. Studies are usually conducted exclusively by researchers from the computer science community, with little to no involvement of social scientists. Works focus on computational aspects such as the usage of modern machine learning techniques to construct and quantitatively evaluate hate speech classifiers. Most annotated datasets are built using crowdsourcing platforms, with annotation guidelines that lack clear definitions of what is considered hate speech.

On the other hand, studies on hate speech from social sciences are mainly of a qualitative nature, such as case studies with detailed analysis. Voluminous information is processed using basic statistical software, not taking advantage of modern machine learning tools.

In this work in progress, we take an interdisciplinary approach to the problem of hate speech in order to detect its appearance in tweets related to newspaper articles about the COVID-19 pandemic. Our goal is to build resources and computational tools to aid in the study of questions from the social field related to hate speech generated during the pandemic.

Our work covers two main aspects. One aspect is the construction of a hate speech corpus, collecting and annotating tweets using the following criteria: 1) based on a thoughtful definition of hate speech, 2) focused on tweets written in Argentinian Spanish, 3) considering context of tweets, and 4) with a clear data statement. The other aspect is the analysis of social media using computational tools in order to answer relevant social questions related to hate speech in the context of the pandemic. For

a preliminary analysis, we present and use a first version of a neural hate speech classifier, trained with currently existing datasets.

The rest of the paper is organized as follows. Section 2 presents previous work. Section 3 describes the corpus construction process. Section 4 presents the questions we aim to answer. It also describes our classifier and a first analysis of the data using it. Finally, Section 5 describes the conclusions reached so far.

## 2 Related work

Hate speech detection is a text classification task related to sentiment analysis. Prior to the dominance of social media, it has been studied for web pages and newsgroups, for instance on the detection of racism and anti-semitism (Greevy and Smeaton, 2004; Warner and Hirschberg, 2012). Later, works started to center on social media, such as MySpace (Thelwall, 2008), Reddit (Saleem et al., 2017) and mainly Twitter.

Several annotated datasets from social media were built, most of them for the English language. Waseem and Hovy (2016) provided a dataset of ∼17k tweets annotated for racism and sexism, later expanded by further work (Waseem, 2016; Gambäck and Sikdar, 2017; Park and Fung, 2017). Davidson et al. (2017) built a ∼24k tweets dataset by crowdsourcing annotations for hate speech and offensive language.

Annotated datasets for Spanish are scarce, despite being one of the three most used languages in social media. To our knowledge, all available datasets were published in the context of shared tasks. Fersini et al. (2018) presented a ∼4k twitter dataset on misogyny for the Automatic Misogyny Identification (AMI) shared task (IberEval 2018). The MEX-A3T task (IberEval 2018 and IberLEF 2019) included a dataset of ∼11k Mexican Spanish tweets annotated for agressiveness (Carmona et al., 2018; Aragón et al., 2019). Basile et al. (2019) published a ∼6.6k tweets dataset annotated for misogyny and xenophobia, in the context of the HatEval challenge (SemEval 2019).

Regarding methods on hate speech detection, approaches range from classic machine learning techniques such as handcrafted features and bags of words over linear classifiers (Waseem and Hovy, 2016; Greevy and Smeaton, 2004; Warner and Hirschberg, 2012), to modern deep learning models that use pretrained embeddings, neural language

| Concept | Quantity |
|---|---|
| original tweets (OTs) | 30,448 |
| news articles (NPAs) | 26,236 |
| tweets in response to original tweets (RPs) | 459,506 |

Table 1: Retrieved dataset statistics.

models and transformers (Gambäck and Sikdar, 2017; Park and Fung, 2017; Badjatiya et al., 2017; Agrawal and Awekar, 2018; Bisht et al., 2020).

On the side of the legal domain, most papers on hate speech are related to its definition and classification, or to the elements that enable the identification of hate speech, and its relationship to freedom of expression and human rights (Torres and Taricco, 2019; CIDH, 2015; Article 19, 2015). However, they do not include any empirical analysis, nor analyze how hate speech could appear in social media.

## 3 Corpus

In this section we describe the corpus building process. For the data collection, we define the data sources and the applied filters. We then define the data statement and our decisions on the data annotation process.

### 3.1 Dataset

To build the dataset, we monitored the official twitter accounts of a selected set of Argentinian newspapers[1] (from now on, the original tweets or OTs) for a fixed period of time.[2] We then scrapped the text of their associated news articles (NPAs) from the newspaper official webpages[3], and collected tweets replies (RPs) to the OTs (e.g. see Fig. 1). Then, we filtered the dataset by selecting only those NPAs (and their corresponding OTs and RPs) containing the following terms: coronavirus, COVID-19, COVID, Wuhan, *cuarentena* (quarantine), *normalidad* (normality), *aislamiento* (isolation), *padecimiento* (suffering), *encierro* (confinement), *fase* (phase), *infectado* (infected), *distanciamiento* (distancing), *fiebre* (fever) and *síntoma* (symptom).

Table 1 shows the statistics of the resulting dataset. There are less NPAs than OTs because news can be referenced by several OTs.

---

[1] La Nación (@LANACION), Infobae (@infobae), Clarín (@clarincom), Crónica (@cronica) and Perfil (@perfilcom).
[2] February 10 to June 9 2020. See Appendix A for detail.
[3] All accessed June 2020.

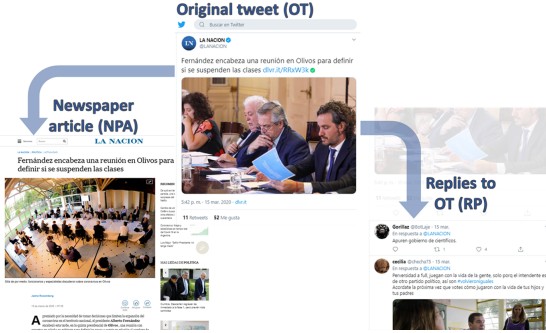

Figure 1: Example illustrating the corpus construction. An original tweet (OT) from newspaper *La Nación*, its associated news article (NPA) and some replying tweets (RPs) from users.

## 3.2 Data Statement

In the following paragraphs we briefly describe the data statement of our corpus, including some annotation decisions. Data statements were proposed in Bender and Friedman (2018) to address critical issues when working with natural language data, such as biases and scope of the data.

Our goal is to obtain Spanish tweets (preferably from Argentina) associated with the COVID-19 pandemic. The gathered tweets are related to country-wide newspapers and not to regional ones. Tweets may, however, be written by inhabitants of different regions of the country or from other countries. In both cases there might be use of regionalisms (Pérez et al., 2019). However, we do not have information of tweet authors' demographics.

The annotation criteria is being developed by a interdisciplinary team of social scientists trained in the detection of discrimination and hate speech and computer scientists with experience developing annotation schemas and criteria. We plan to have a team of three annotators with background in social science studies and whose mother tongue is Spanish. Inter-annotator agreement will be measured.

## 3.3 Annotation schema and criteria

We are currently working on the definition of the annotation schema and criteria. We took the following decisions up to the moment:

1) We will annotate user-generated tweets replying to original tweets (RPs).

2) The following aspects will be annotated: a) Is hate speech present in the tweet? The presence will be determined according to a definition based on the ones presented by the Center for Studies on Freedom of Expression and Access to Infor-

mation (CELE)[4] (Torres and Taricco, 2019) and by the Inter-American Commission on Human Rights (CIDH)[5] (CIDH, 2015)). b) If hate speech is present, which is the discriminative motive? As motives, a relevant subset of those provided by the Argentinian National Institute against Discrimination, Xenophobia and Racism (INADI)[6] has been chosen:

- violence against women
- gender identity or sexual orientation
- racism or xenophobia
- poverty, socioeconomic situation or neighbourhood of residence
- religion
- disability or mental health

c) If it contains hate speech, does it call for action against the group or a member of the group?

3) We select a subset of NPA containing some "seed" words (for instance *China, Cuba, Bolivia, feminist, jails, names of actresses, or well-known women, etc.*) that might indicate topics triggering hate speech. After this, we perform a manual selection of this subset looking at both the content and comments.

4) To avoid sparsity, we consider only articles that have twenty or more RPs. Of those, we randomly select a fraction of RPs to be annotated. RPs for the same OT will be assigned to the same annotator.

5) In the process of annotation, we bring the tweet (the RP) and the original tweet (OT) to bring annotators the context of the comment

We are currently working on the development of the annotation tool and the selection process of the annotators. The annotation guidelines development process will be similar to the MAMA portion of the MATTER cycle (Pustejovsky and Stubbs, 2012).

## 4 Analysis

In this section we describe the questions we want to answer by the analysis of the collected data. We also describe our current hate speech classifier, and some preliminary results obtained using it.

---

[4]CELE: https://www.palermo.edu/cele/. Accessed June 2020.
[5]CIDH: http://www.oas.org/es/cidh/. Accessed June 2020.
[6]INADI: https://www.argentina.gob.ar/inadi. Accessed June 2020.

## 4.1 Questions

Our goal is to understand the relationship among hate speech in social media and newspaper articles in the context of the COVID-19 pandemic. There are two lines of work: one of a descriptive nature and the other of an explanatory nature (Babbie, 2015). In the first one, our aim is to characterize those textual elements that configure hate speech in the context of the current pandemic. We identify the following questions: 1) is there a continuity between hate speech during the pandemic and those that previously existed?, 2) is there any difference in how hate speech is expressed by users in the different newspapers and over time?, and 3) which communities are targeted by hate speech in the pandemic context? In the explanatory dimension we want to determine which factors affect the emergence of hate speech in reply to news replicated on social networks. On this line, our questions are: 1) to what extent do newspaper articles induce the emergence of hate speech?, 2) is hate speech linked to a *snowball* effect or to the performance of some *influencer* users?, and 3) is there a link or community among people who produce hate speech?

## 4.2 Hate Speech Classifier

For a preliminary analysis we developed a first version of a hate speech classifier. We based our classifier on BETO (Canete et al., 2020), a pre-trained version for the Spanish language of the general-purpose neural model BERT (Devlin et al., 2019). A linear layer was used on top of BETO to compute a final hatefulness score for the input text. A threshold hyperparameter was used to decide the classification.

The classifier was trained and evaluated using the Spanish dataset for the HatEval challenge (SemEval 2019) oriented at the detection of misogyny and xenophobia (Basile et al., 2019). This dataset contains 6,600 tweets from different spanish speaking countries (mainly from Spain). The evaluation on the test set gives an $F_1$-score of 0.75, a result above the best systems presented in the challenge.

In this first version, no further training, fine-tuning or domain adaptation was done to accommodate the nature of our new dataset, where tweets are written mainly in Argentinian Spanish and hate topics are different. So, it is to be expected that the classifier does not perform as good as in the original domain.

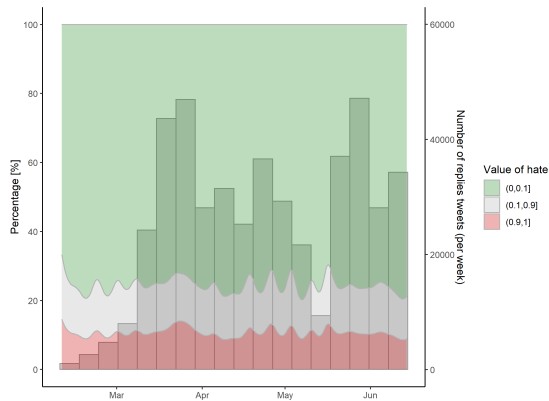

Figure 2: Hate speech distribution over time and histogram with number of tweets for each week. Tweets are divided in three regions according to hate score. Instances with the biggest hate scores, in range 0.9 to 1, are at the bottom region (red).

## 4.3 Preliminary Results

Using our hate speech classifier, we computed hate speech scores for all the RP tweets in our dataset. The threshold was adjusted by hand to 0.9, in order to reduce the observed bias towards the positive class (hateful tweets). With this setting, we found that 9% of the tweets contain hate speech. Distribution of hate speech over time can be seen in Fig. 2. Here, we find a pattern that shows peaks of hate speech on weekends. Further analysis is required to explain this pattern.

## 5 Discussion

In this short paper we presented work in progress on hate speech in social media during the COVID-19 pandemic. In our work we take an interdisciplinary approach, covering aspects related to both computer and social sciences. This approach is challenging but also advantageous.

The construction of our corpus will allow the training and evaluation of new hate speech classifiers. These classifiers will be useful not only for detection, but also for a better analysis and understanding of the hate speech phenomena.

## Acknowledgments

This work was partially funded by *Programa Interdisciplinario de la UBA sobre Marginaciones Sociales* -Interdisciplinary Program on Social Marginalizations- (PIUBAMAS RESCS-2019-1913-E-UBA-REC, Universidad de Buenos Aires, Argentina).

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

## A   Appendix

The date 02/10/20 was chosen as the start date for the retrieval of tweets for many reasons. In this date: 1) a team of the international World Health Organization (WHO) arrived to China to help contain the transmission of the virus, 2) the number of 1,000 deaths worldwide was reached and from this day on more than 100 daily deaths were reported in China, 3) there were more than 440 confirmed cases in more than 25 countries and territories outside of mainland China -distributed in 4 of the 5 continents-, 4) A day earlier, COVID had exceeded the death toll from SARS. 5) from this day on the number of detected cases began to grow rapidly outside China, and finally, 6) one day after the WHO defined COVID-19 as a new illness.[7]

---

[7] https://www.who.int/emergencies/diseases/novel-coronavirus-2019/situation-reports/.