# OpenReview forum: "A study of Hate Speech in Social Media during the COVID-19 outbreak"
_EMNLP/2020/Workshop/NLP-COVID — Submitted to NLP-COVID19-EMNLP_

### Official Review · AnonReviewer2 · 2020-09-09
**A research proposal with interesting research questions that need to be fleshed out**

**Rating:** 4
**Confidence:** 4

**Review:**

This work proposes to annotate Spanish-language tweets, associated with coronavirus news articles, for hate speech. The article discusses the intended plan to annotate a set of tweets using multiple annotators and then build a machine learning system to identify more tweets containing hateful language. These tweets would then be used to answer several questions about hate speech during the coronavirus pandemic. Finally, the paper presents preliminary results from a separate machine learning system.

Firstly, I find the article clearly written and easy to understand. This work appears to be primarily a research proposal with a description of the proposed steps and an outline of the research questions to be discussed. The data collection description is fairly clear, but the intended questions to ask are general and not well fleshed out. Unfortunately the additional data analysis (in sections 4.2 and 4.3) does not link well with the proposal to provide any support or specific research direction. While the call for papers for this workshop mentions work in progress, this article seems to be a little too early work in progress.

Some further issues with the paper are outlined below
- The paper mentions Bender & Friedman’s data statements but does not provide the long form of a data statement (or as close as possible at this stage) as suggested by the cited paper
- Apart from one research question in Section 4.1 (discussed below), the paper doesn’t provide reasoning why hate speech should be studied specifically during the coronavirus pandemic as opposed to any other time.
- The data analysis in the final section sounds quite interesting. You state that your results outperform the current best methods for the SemEval 2019 HatEval challenge. Oddly, you then claim that it “does not perform as good as in the original domain”, Maybe I’m misunderstanding, but that performance is good, right?
- Section 4.1 provides a brief overview of the types of questions that you intend to ask with the final Twitter dataset. This is really lacking detail and it’s unclear how you will answer several of these questions with the dataset that you describe. For the first descriptive question, how will you evaluate trends of hate speech from before the pandemic when your dataset will only include tweets on coronavirus articles? For the first explanatory question, how will evaluate the effect of newspaper articles on hate speech if you only have tweets related to newspaper articles and not other sources? You also suggest doing some sort of network analysis on Twitter users to identify influencers and community but don’t provide any details. Furthermore it is not clear if the data collected will contain this information.
- There’s a strange dip in tweets in mid May in Figure 2. Is there an interesting reason behind that?

---

### Official Review · AnonReviewer3 · 2020-09-10
**Good preliminary work but needs more before publication**

**Rating:** 3
**Confidence:** 3

**Review:**

I think this work might make for a good paper if the dataset annotation had been completed and there was a comprehensive analysis of the data. Overall, I agree with the sentiment of Reviewer 2 in that the direction and questions are interesting but a more comprehensive result if needed for publication.

I think figure 2 could be seen as a contribution if the authors had explained why the findings that they purport are significant and what question they answer. Moreover, the questions mentioned in section 4.1 are important and interesting but do not seem related to the preliminary findings they present in the paper.

Once the dataset is collected, the authors should describe the dataset in detail and provide examples to help readers understand the value of their collected dataset.

I very much look forward to seeing a completed version of this work the analysis of results.

---

### Official Review · AnonReviewer1 · 2020-09-25
**Great idea but with some flaws in the experiment description and comprehensibility of the results.**

**Rating:** 4
**Confidence:** 5

**Review:**

The paper describes work in progress whose goal is to the collect a hate speech corpus with Argentinian Spanish Tweets about COVID-19. Once the collection is finished, authors also intend to answer questions regarding social science, using SOTA computational tools. Some of these are:

-  is there a continuity between hate speech during the pandemic and those that previously existed?
- is there any difference in how hate speech is expressed by users in the different newspapers and over time?
- which communities are targeted by hate speech in the pandemic context?
- to what extent do newspaper articles induce the emergence of hate speech?
- is hate speech linked to a snowball effect or to the performance of some influencer users?
- is there a link or community among people who produce hate speech?

Although the idea is great and the work is going on a very positive direction, I saw some flaws in the description of the experiment. Moreover, despite the fact this is a short paper, I agree with the reviewers that more comprehensive results are necessary before the publication of the manuscript/

Here are some suggestions to the authors:

1. The acronyms OT, NPAs and RPs are counter-intuitive and confusing. I suggest the authors to change them or refer to the proper term. The fluency of the paper will increase.
2. How were the query search (COVID-19, COVID, Wuhan…) and seed words (China, Bolivia, etc.) defined?
3. If one of the paper’s goal is to obtain Spanish tweets (preferably from Argentina) associated with the COVID- 19 pandemic, why haven’t you used the geolocation function from Twitter? (However, we do not have information of tweet authors’ demographics. )
4. How many data was annotated so far?